# The Scale Effect of Double-Ring Infiltration and Soil Infiltration Zoning in a Semi-Arid Steppe

**Mingyang Li**[ORCID]**, Tingxi Liu \*, Limin Duan, Yanyun Luo, Long Ma \*, Junyi Zhang, Yajun Zhou and Zexun Chen**

Inner Mongolia Water Resource Protection and Utilization Key Laboratory, College of Water Conservancy and Civil Engineering, Inner Mongolia Agricultural University, Hohhot 010018, China
\* Correspondence: txliu@imau.edu.cn (T.L.); malong4444333@163.com (L.M.); Tel./Fax: +86-471-4309386 (T.L.)

**Abstract:** The double-ring infiltrometer is widely used to measure soil unsaturated hydraulic conductivity in the field. The scale effect of the inner and outer ring size (especially the inner one) affects the measurement results. In the semi-arid steppe, where water is scarce and transportation is inadequate, studying the scale effect caused by the inner-ring diameter of the infiltrometer can reduce the test consumption on the premise of ensuring the test accuracy. In this paper, a total of 190 double-ring infiltration tests with different inner-ring diameters (15, 20, 25, 30, and 40 cm) and 0.33 times outer buffer index were carried out at 38 sites with five soil types in the Xilin river basin, China. Results showed that: (1) When comparing the simulated parameters of six infiltration models, parameters increased with the increase of the infiltrometer inner diameter, but the trend gradually slowed down, indicating that the increase of the infiltrometer inner diameter would weaken the influence of the infiltrometer scale effect. However, the infiltrometer with an inner diameter of 40 cm is not enough to completely overcome the scale effect. (2) Through principal component analysis, the infiltration process is mainly affected by the particle size and the initial moisture content. (3) The soil infiltration map based on infiltration tests was more practical than the soil type map, which can provide a theoretical basis for ecological and soil restoration in the future.

**Keywords:** soil water-holding capacity; infiltrometer diameter; infiltration models; soil restoration; Xilin River basin

---

## 1. Introduction

Against the background of global climate change, the intensity and frequency of human activities are continuously increasing, and the natural ecosystem is being severely damaged [1–3]. For example, severe ecological deterioration has occurred in both Hooghly-Matla, India [4], and Heihe basins, China [5]. The semi-arid grasslands in the central and western parts of China, have suffered serious degradation [6]. Previous studies showed that the primary productivity in arid and semi-arid regions of Inner Mongolia, China, had significantly decreased since 2000 [7], with some areas even showing desertification [8].

There have been relatively few successful international cases attempting to control degraded ecosystems. For instance, papyrus, a biological element, is being used to control and repair lakes Naivasha and Victoria in East Africa [9]. Soil amendment with biochar can result in decreased bulk density and soil penetration resistance, and increased water-holding capacity [10]. Muñoz-Rojas et al. [11] assessed the functionality of restored soils in degraded semiarid ecosystems, which showed a positive effect of vegetation on reconstructed soils. Indeed, soil quality is critical to ecosystem restoration, and the water-holding capacity of soil in arid areas is an important indicators of soil quality. Good water-holding capacity contributes to the growth of plant roots [12], thus improved

species diversity does not reduce, but rather improves, the soil erosion resistance [13,14], and finally allows the whole ecosystem to achieve a positive cycle [15].

To achieve the goal of obtaining the soil water retention characteristics of the study area, it is necessary to accurately evaluate this capacity, which can be estimated by the final infiltration rate (FIR) and infiltration time when the infiltration rate first reaches the final infiltration time (FIT). A slower infiltration rate indicates that the soil has better water retention [10]. The process of soil infiltration can be simulated in the laboratory using the ring-knife or a soil column [16,17]. However, such studies are influenced by disturbances of the original soil bulk density, porosity, and other soil physical properties. Instead, an infiltration test based on in situ undisturbed soil is capable of reflecting the original state of soil more effectively.

Compared with a single-ring infiltrometer, a double-ring infiltrometer can be used to construct an infiltration buffer, which reduces the main external interference factors on the infiltration process of the inner ring [18]. Previous studies have shown that the average infiltration rate increases with the infiltrometer diameter [19,20]. Lai et al. [21] found through a large number of simulations that, compared with the outer buffering index (*bi*), the inner-ring diameter (*di*) plays a more important role in obtaining a relatively stable and representative measurement result. When the *bi* is equal or larger than 0.33, the buffer may meet the measurement requirements.

The study area of the paper was located in the Xilin river basin, Inner Mongolia, China, and was in the grazing district [6,22]. The local soil type map and available data and experiments are not detailed enough [23] to display the area soil water retention. Thus, it is necessary to reclassify the division map of soil water retention based on the double-ring infiltration test to provide an important theoretical basis for later ecological governance [24].

In this study, 38 soil double-ring infiltration test sites were arranged according to the proportion of different soil types in the study area. Five kinds of double-ring infiltrometers with different inner-ring diameters and same outer buffering index (*di* = 15, 20, 25, 30, and 40 cm, *bi* = 0.33) were set up for each test site, and six widely used field soil infiltration models were used for simulating five soil infiltration processes for five soil types. The simulated model parameters were compared to investigate the scale effect of double-ring infiltration. The final infiltration rate (FIR) and final infiltration time (FIT) of different sampling sites, calculated using the model with good performances, were divided into three groups using k-means clustering. Based on these groups and the results of principal component analysis (PCA), we obtained the soil infiltration zones and the influencing factors of the infiltration process map. This study explores and reveals the special ecological hydrological processes and water conversion and transport mechanisms and their influencing factors in grassland basins in arid and semi-arid regions, which is significant for further protection of the ecological environment and the treatment of degraded pastures.

## 2. Materials and Methods

### 2.1. Experimental Sites

The semi-arid steppe Xilin River basin located in Xilinhaote City, Xilingol, Inner Mongolia, China, was selected. Field trials were carried out at the junction of the Holertugol, a tributary of Xilin River, and the Xilingol, the main stream, located at approximately 43°24″–44°4″ N, 116°17″–117°15″ E (Figure 1a). With an area of 1852 km², the topography is surrounded by mountains on three sides. More than 90% of the vegetation is natural forage grass, with *Leymus chinensis* being the most common. The study area belonged to the natural pasture and has a semi-arid continental monsoon climate in the middle temperate zone, with an average annual precipitation of 266.8 mm. The precipitation in June and August accounts for more than 50% of the annual total.

The study area contained thick chestnut loess (TCL), meadow swamp soil (MSS), desert aeolian soil (DAS), limy meadow sandy soil (LMSS), and pale black soil (PBS), for a total of five soil types (Figure 1b). The variation of soil parameters from 0–50 cm deep relative to soil type are presented

in Table 1. With the low clay content, the soil sand content was extremely high, and its average content was more than 80%. MSS and LMSS were distributed on both sides of the river, and the higher water content nurtured a large number of wetland vegetation, making their soil organic matter and underground biomass higher than the other three kinds of soil [25,26].

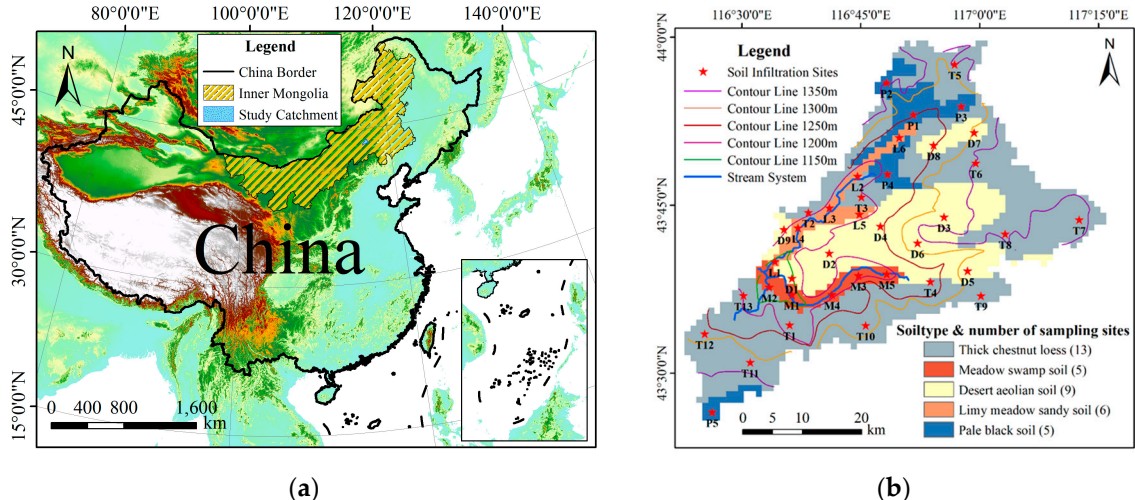

**Figure 1.** (**a**) Location of the study catchment within Inner Mongolia, China; and (**b**) soil types of the study catchment as well as the sampling sites used in this study.

**Table 1.** Variation of soil parameters relative to soil type in Xilin River Basin from 0–50 cm.

| Soil Type | Sand (%) | Silt (%) | APS (µm) | BD (g cm$^{-3}$) | OMC (g kg$^{-1}$) | IMC (%) | UB (kg m$^{-3}$) |
|---|---|---|---|---|---|---|---|
| TCL | 84.013 | 14.601 | 85.291 | 1.621 | 1.923 | 5.148 | 1.229 |
| MSS | 85.573 | 12.946 | 86.871 | 1.572 | 2.989 | 26.600 | 3.468 |
| DAS | 92.384 | 7.226 | 93.765 | 1.466 | 0.577 | 8.754 | 1.459 |
| LMSS | 77.310 | 20.909 | 78.511 | 1.607 | 2.318 | 17.542 | 1.975 |
| PBS | 82.356 | 16.568 | 83.615 | 1.556 | 1.708 | 4.218 | 1.766 |

**Note:** In the table, Sand stands for sand content, Silt for silt content, APS for average particle size, BD for soil bulk density, OMC for soil organic matter content, IMC for initial water content, and UB for underground biomass.

### 2.2. Experimental Design and Data Acquisition

According to the 1:1 million Chinese soil type distribution map (http://westdc.westgis.ac.cn) and the proportion of each soil type, several infiltration sites were set up. Following dry and clear weather in the first three days, the double-ring infiltration test was carried out on July 2018 using a hand-held GPS positioner after removing the surface floating soil and vegetation, which avoided inconvenience in measurement while not affecting the results and no precipitation during the test.

Five infiltrometers with different diameters and a height of 50 cm were used to measure the soil infiltration process. The inner-ring diameters were 15, 20, 25, 30, and 40 cm, and the *bi* were 0.33, which means the outer-ring diameter was 1.33 times that of the inner-ring. To keep the water supply at the same level constant, two specially designed 25 L Mariotte bottles were connected to the inner and outer infiltrometer rings. A small amount of pigment was added to observe the changes of the water level (Figure 2). Before the infiltration, cutting rings were used to collect soil at 10 cm layer intervals from 0 to 50 cm at each infiltration site for measuring the soil particle size, bulk density, organic matter content, initial soil moisture content, and underground biomass. In the process of infiltration, the changes of the water level of the Mariotte bottles and the inner and outer infiltrometer rings were recorded every minute in the first 20 min and every 3 min from 20 to 80 min. After the experiment, the infiltrometers were pulled out as quickly as possible to prevent redundant water infiltration. The section at the center of the infiltrometer was excavated to observe the infiltration depth.

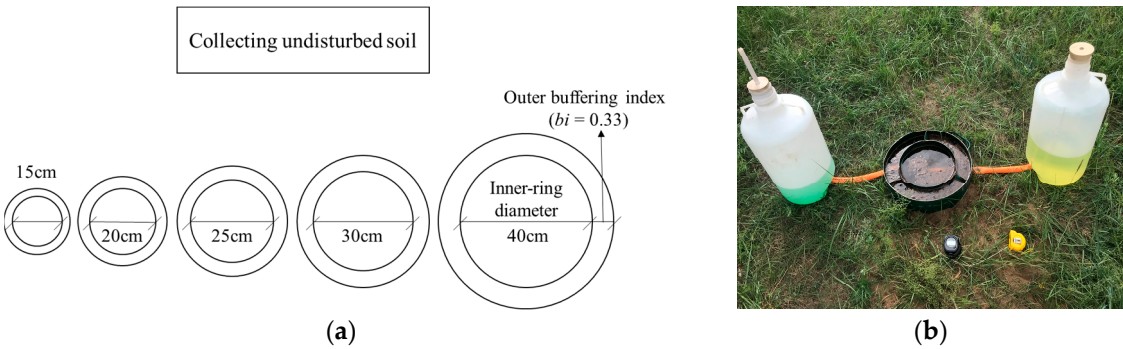

**Figure 2.** (**a**) Schematic diagram of double-ring infiltration test at each sampling site, and (**b**) experimental equipment.

The soil particle size was measured using the dry method by the Helos & Rodos laser particle size analyzer (Sympatec GmbH, Clausthal, Germany). The size classification standard that was adopted was the USDA classification system of soil texture: clay (<0.002 mm), silt (0.05 mm–0.002 mm), and sand (2 mm–0.05 mm) [27]. Bulk density was measured using a 100 cm$^3$ cutting ring. Organic matter content was measured using the treatment of concentrated sulfuric acid with potassium dichromate and external heating. Initial soil moisture content was measured using the constant temperature incubator drying method. Underground biomass was measured using the dry weighing method. Additionally, soil particle size, bulk density, organic matter content, initial soil moisture content, and underground biomass were the average of the entire 0–50 cm depth for three replicates at each site.

*2.3. Infiltration Models and Data Analysis*

2.3.1. Infiltration Models

Soil infiltration models can simulate the relationship between infiltration time and cumulative infiltration. The following six infiltration models were selected to simulate the double-ring infiltration process of five infiltrometers with different diameters [27–32].

2.3.2. Model Evaluation and Data Analysis

All model parameters were fitted using the curve fitting tool in MATLAB R2016b (The MathWorks, Natick, MA, USA), where the nonlinear regression required information such as the form of the infiltration model, the threshold of the parameters, and the number of iterations. The choice of the best parameters was determined by using the least squares method, a mathematical optimization technique, by calculating the least squares difference of the different parameters. The simulated parameters were used as inputs for the six infiltration models selected to calculate the cumulative infiltration amount with the same recording time as the measured data. The performance of each infiltration model was evaluated by comparing the measured and predicted cumulative infiltration.

The selected soil-infiltration models were nonlinear. Therefore, the *Adj-R*$^2$ was used instead of the coefficient of determination ($R^2$), and consideration was given to the results of the extra variables [33]. Model efficiency (*NSE*) was also used. *NSE*, introduced by Nash and Sutcliffe [34], is similar to the correlation coefficient from linear regression, $R^2$. However, an important difference between the *NSE* and $R^2$ values is that the *NSE* compares the predicted values to the 1:1 line between the measured and predicted values rather than the best regression line through the points [35]. Note that the *NSE* can be negative.

One-way ANOVA was used to compare the average estimated parameters of each simulated infiltration process at the level of $p < 0.05$. The reduced chi-square is equivalent to the residual sum of squares of the one-way ANOVA. This value is actually the error between the fitted values and the actual

values, and smaller values indicate stronger degrees of curve fitting [33,36]. The root-mean-square error (RMSE) was used to measure the deviation between the simulated and the measured value [37].

*Adj-R²*, *NSE*, the reduced chi-square, and RMSE are defined as follows:

$$Adj - R^2 = 1 - \frac{RSS/dfE}{TSS/dfE} \tag{1}$$

$$NSE = 1 - \frac{\sum_{t=1}^{n}\left(I_o(t) - I_p(t)\right)^2}{\sum_{t=1}^{n}\left(I_o(t) - \overline{I_o}\right)^2} \tag{2}$$

$$\overline{\chi^2} = \frac{\chi^2}{dfE} = \frac{RSS}{dfE} \tag{3}$$

$$RMSE = \sqrt{\frac{\sum_{t=1}^{n}\left(I_p(t) - I_o(t)\right)^2}{n}} \tag{4}$$

where $\chi^2$ is the reduced chi-square; *RSS* is the residual sum of squares; *TSS* is the total sum of squares; *dfE* is the error degrees of freedom; $I_o(t)$ and $I_p(t)$ are the observed and predicted cumulative infiltration (cm) at infiltration time *t*, respectively; and *n* is the total number of measurement times.

K-means clustering, which is popular for cluster analysis in data mining, aims to partition n observations into k clusters in which each observation belongs to the cluster with the nearest mean, serving as a prototype of the cluster [38]. This results in a partitioning of the data space into Voronoi cells. Accordingly, cluster centers were used to model the FIR and the FIT data into three parts, which could be used to divide the study area into different comparable soil infiltration spatial extents.

PCA uses an orthogonal transformation to convert a set of measured soil properties of possibly correlated variables into a set of values of linearly uncorrelated variables called principal components [37]. Then, a correlation matrix (equivalent to the number of soil properties of each sampling site), and then eigenvectors (coefficients of components) and eigenvalues (variance of eigenvectors) were calculated. From these, three infiltration parts could be easily distinguished by eigenvectors, and measured soil properties could be reduced to two principal components using PCA, which explained at least 70% total variance in the original dataset.

## 3. Results and Discussion

### 3.1. Infiltration Process

The initial infiltration rate was very fast, with an average of 7.2 cm/min per unit. The infiltration rate and cumulated infiltration were directly proportional to the infiltrometer area, and the infiltration rate per unit area was independent of the infiltrometer diameter (Figure 3). The initial infiltration rate and the cumulated infiltration of the five infiltrometers with increasing diameters in 80 min were 1.36, 2.51, 3.98, 5.75, and 10.39 cm/min, and 6.23, 11.87, 18.90, 27.29, and 50.20 L, respectively. Exceptionally, both the initial infiltration rate and cumulated infiltration of the 15 cm inner diameter infiltrometer were lower than the estimated level in proportion to area.

For the whole study area, the soil water infiltration rate was very fast. The average FIR per unit area was up to 0.17 cm/min. At the early stage of infiltration, the infiltration rate declined sharply, reaching 79.6% and 95.3% in the first 5 and 20 min, respectively. The whole infiltration process was stable and fast, and the FIT was achieved within about 20–60 min, which indicates that the vertical structure and composition of this sandy meadow soil was relatively stable.

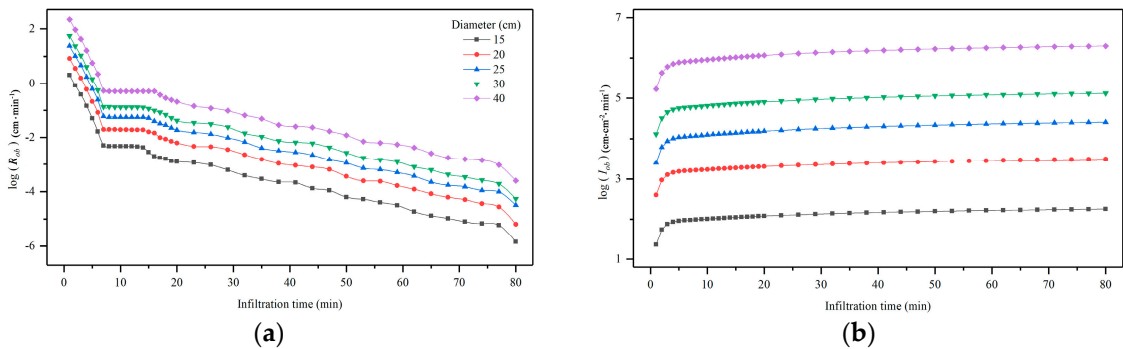

**Figure 3.** The log-transformed measured infiltration rate (**a**) and cumulative infiltration (**b**) with different inner-ring diameter and the same outer buffer index (*di* = 15, 20, 25, 30 and 40 cm, *bi* = 0.33). $R_{ob}$ and $I_{ob}$ are the infiltration rate and cumulative infiltration of different inner-ring diameters, respectively.

## 3.2. Model Parameters and Performance

After unifying the double-ring infiltration processes of five infiltrometer diameters to the unit area, the parameters were calculated using the models in Table 2. The parameters $s_m$, $S$, $A^*$, $\beta$, $f_c$, $k$, $\beta'$, and $S$ basically showed an increasing trend with an increase of the inner-ring diameter, and the increasing trend gradually slowed down (Figure 4). This indicates that the double-ring infiltration scale effect gradually weakened with the increase of the inner-ring diameter, and even though the largest inner-ring (*di* = 40 cm) still had a scale effect, it was significantly improved compared with the other four, which is consistent with the research results of Lai et al. [21].

**Table 2.** Approximate theory-based and empirical infiltration equations.

| Model Type | Model Name | Equation | Parameters |
|---|---|---|---|
| Theoretical models | Green–Ampt (1911) [28] | $I(t) = K\frac{H_a + s_m + z}{z}$ | $K$ is the saturated hydraulic conductivity of the transmission zone (cm/min), $H_a$ is the thickness of surface water (cm), $s_m$ is the average potential suction of the wetting front (cm), and $z$ is forward distance of the wetting front (cm). |
| | Philip (1954) [29] | $I(t) = \frac{1}{2}St^{-0.5} + A$ | $S$ is the sorptivity (cm·min$^{-0.5}$) and $A$ is the transmissivity factor (cm/min). |
| Empirical models | Kostiakov (1932) [30] | $I(t) = \alpha t^{\beta}$ | $\alpha > 0$ and $0 < \beta < 1$ are dimensionless empirical constants. |
| | Horton (1940) [31] | $I(t) =$ $f_c t + \frac{1}{k}(f_c - f_0)(1 - e^{-kt})$ | $f_0$ and $f_c$ are the presumed initial and final infiltration rates; $k$ is a constant that determines the rate at which $f_0$ approaches $f_c$. |
| | Mezencev (1948) [32] | $I(t) = K't + \alpha't^{\beta'}$ | $K' > 0$, $\alpha' > 0$, and $0 < \beta' < 1$ are dimensionless empirical constants. |
| | USDA-NRCS (2003) [27] | $I(t) = at^{-b} + 0.6985$ | $a$ and $b$ are dimensionless empirical constants. |

Note: $I(t)$ is the cumulative infiltration (cm) and $t$ is the infiltration time (min).

Comparing the infiltration for different soil types, it was found that the DAS infiltration rate was the fastest, and the model parameters' values for this soil type were also large. The MSS infiltration rate was the slowest, and the model parameters were all small. In the case of the same inner-ring diameter, the values of Kostiakov's $\alpha$ and Kostiacov-Lewis's $\alpha'$ were very similar to the value of USDA-NRCS's $a$, which had the same magnitude order as Philip's $S$ (sorptivity). This implied that these three model parameters had the same physical meaning as $S$ [33]. Based on the Figure 4, the parameters $\beta$, $f_c$, $k$, $\beta'$, and $b$ were more stable for the same soil type. That is to say, these model parameters were indices that represented the permeability characteristics, similar to the report by Shukla et al. [39].

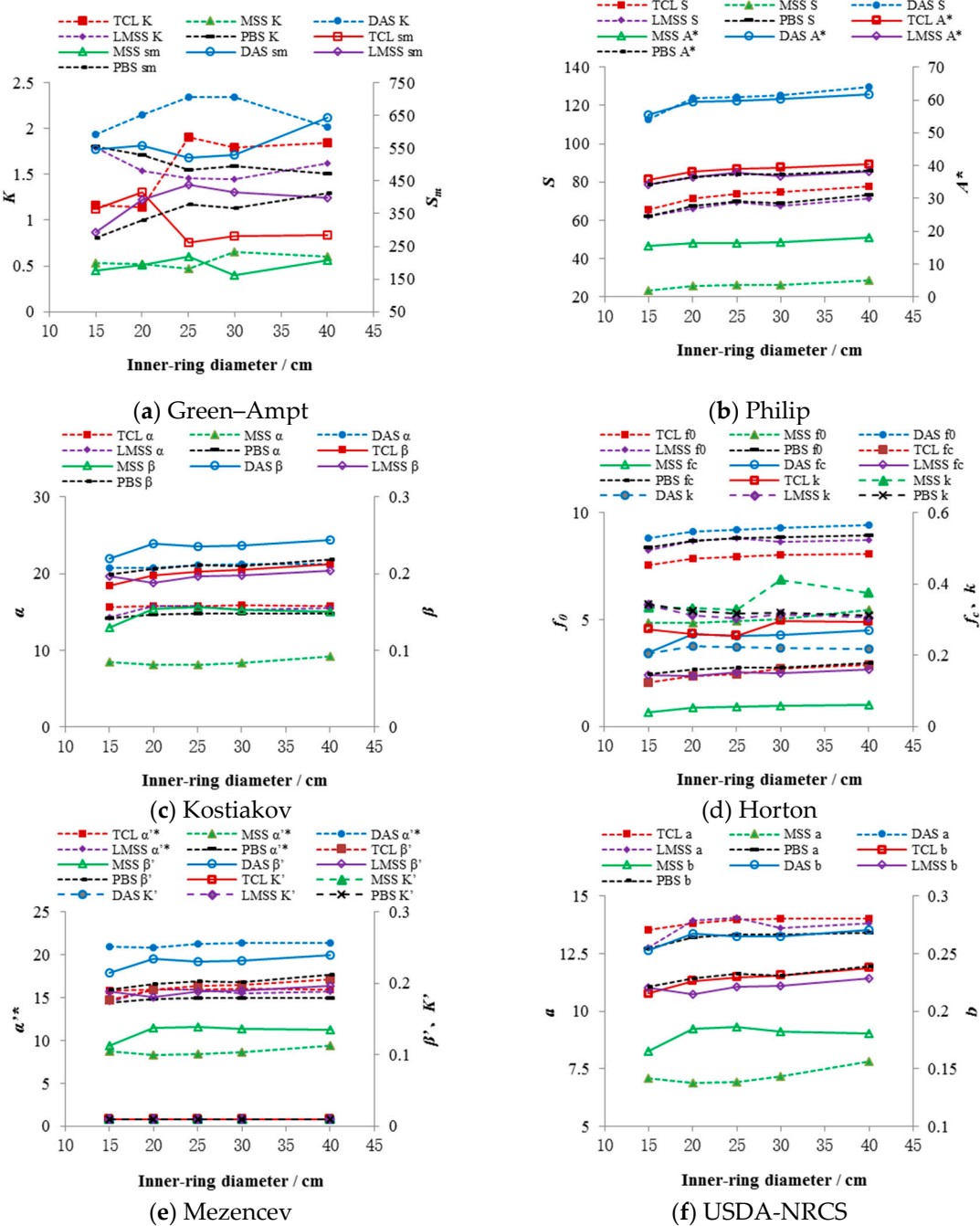

**Figure 4.** Model parameters of the six infiltration models with five kinds of infiltrometer diameters for five soil types in Xilin River Basin (TCL, MSS, DAS, LMSS, PBS represent thick chestnut loess, meadow swamp soil, desert aeolian soil, limy meadow sandy soil, and pale black soil, respectively).

With the same inner-ring diameter, the simulated cumulative infiltration using the Green–Ampt model was low in the early and medium infiltration periods, and the later simulation value was high. Philip's model had a low estimate in the late stages of the infiltration process, which converged faster when infiltration time (*t*) was small. When the *t* value was increasing, the result of the vertical infiltration of Philip's model was not in effect. This is consistent with Touma's conclusion [40]. The Kostiakov, Mezencev, and USDA-NRCS models had a certain degree of high permeability in the early stage of infiltration. However, the Horton model, which considers the initial infiltration rate, had the best simulation result in four empirical models at the early stage of the infiltration process. Compared

with the same soil type, the cumulative infiltration per unit area increased slightly with the increase of the inner-ring diameter.

The accuracy and error of each model using different sizes of infiltrometers are described in Table 3. The combination of *Adj-R²* and NSE made up for the shortcomings of estimating the accuracy of the models. Although the *Adj-R²* of the Green–Ampt model was high, the *NSE* value was negative, indicating that the effect of this model was extremely poor. Overall, four empirical models were relatively highly accurate, where *Adj-R²* was basically greater than 0.9, which is consistent with the simulation effects of long-term wastewater irrigation using empirical infiltration models in a semi-arid region, as seen in Razzaghi et al. [41]. The results of the one-way ANOVA showed that, except for the Green–Ampt model, the error of each group was small, indicating that the change of test environment had little influence on the test results, and the double-ring infiltrometers made in this study had universality. Both the reduced chi-square and RMSE values decreased with the decrease of the inner-ring diameter. The Horton models were the best quantified and the results were similar to those of Wang et al. [33] and Babaei et al. [42].

**Table 3.** Four evaluation index values of the six evaluated soil infiltration models for five different inner-diameter infiltrometers.

| Evaluation Index | Diameter (cm) | Green–Ampt | Philip | Kostiakov | Horton | Mezencev | USDA-NRCS |
|---|---|---|---|---|---|---|---|
| *Adj-R²* | 15 | 0.894 * | 0.840 * | 0.894 * | 0.974 ** | 0.884 ** | 0.891 ** |
| | 20 | 0.907 ** | 0.852 * | 0.907 ** | 0.975 ** | 0.899 ** | 0.904 ** |
| | 25 | 0.909 ** | 0.852 * | 0.909 ** | 0.975 ** | 0.903 ** | 0.907 ** |
| | 30 | 0.910 ** | 0.865 * | 0.910 ** | 0.976 ** | 0.902 ** | 0.907 ** |
| | 40 | 0.913 ** | 0.866 * | 0.913 ** | 0.976 ** | 0.907 ** | 0.911 ** |
| *NSE* | 15 | −2.110 | 0.945 | 0.979 | 0.960 | 0.973 | 0.955 |
| | 20 | −1.514 | 0.948 | 0.981 | 0.963 | 0.977 | 0.963 |
| | 25 | −1.455 | 0.950 | 0.981 | 0.964 | 0.978 | 0.965 |
| | 30 | −1.365 | 0.951 | 0.982 | 0.965 | 0.976 | 0.963 |
| | 40 | −1.248 | 0.960 | 0.982 | 0.966 | 0.979 | 0.967 |
| Reduced Chi-Square | 15 | 1.031 | 0.034 | 0.004 | 0.004 | 0.004 | 0.005 |
| | 20 | 1.027 | 0.030 | 0.004 | 0.003 | 0.004 | 0.005 |
| | 25 | 1.026 | 0.028 | 0.004 | 0.003 | 0.004 | 0.005 |
| | 30 | 1.027 | 0.028 | 0.004 | 0.003 | 0.004 | 0.005 |
| | 40 | 1.025 | 0.024 | 0.004 | 0.002 | 0.004 | 0.005 |
| RMSE | 15 | 1.190 | 0.173 | 0.064 | 0.052 | 0.066 | 0.073 |
| | 20 | 1.178 | 0.161 | 0.065 | 0.051 | 0.066 | 0.072 |
| | 25 | 1.174 | 0.160 | 0.064 | 0.051 | 0.066 | 0.072 |
| | 30 | 1.175 | 0.153 | 0.064 | 0.049 | 0.066 | 0.072 |
| | 40 | 1.171 | 0.148 | 0.064 | 0.048 | 0.066 | 0.071 |

Note: * indicates that the trend of increasing or decreasing is significant when $\alpha \leq 0.05$, ** indicates that the trend of increasing or decreasing is significant when $\alpha \leq 0.01$.

The overall simulation performance of the Green–Ampt model in the study catchment was poor, and the reasons for this result may be as follows: (1) The wet front did not advance vertically, which is similar to the view of Clemmens and Bautista [43]. (2) The derived implicit function of infiltration time *t* and infiltration amount *I(t)* was not suitable for parameter simulation with the MATLAB curve fitting tool. Philip's model had a bad performance in the late infiltration stage when the simulation value was relatively small. The study speculated that the model might not be applicable for simulating soil with a high sand content, where the infiltration rate decreases rapidly, which was the same as Touma's conjecture regarding the comparison of the soil hydraulic conductivity by the equation of van Genuchten and other capillary models [40]. Although the Kostiakov model performed well in the precision and error analysis, due to the form of its model, as the infiltration time tends to infinity, the

infiltration rate tends to 0, which means that it is recommended to be used in simulations where the infiltration process occurs over a short time.

### 3.3. Difference Analysis of Soil Type and Soil Infiltration Process

Considering the accuracy and error of the models, Horton's model was selected to simulate the FIR at each site. Although the $f_c$ of Horton's model is an empirical simulation value, it is close to a real FIR [44]. Similar to the variation trend of model parameters, the FIR at each site also showed a trend of slow increase with the increase of inner-ring diameter, and the increasing trend gradually slowed down (Figure 5). TCL, MSS, and DAS had similar infiltration sites that produced a log-formed FIR of around $-3$ cm·min$^{-1}$, indicating that there were similar infiltration processes in the above three soil types.

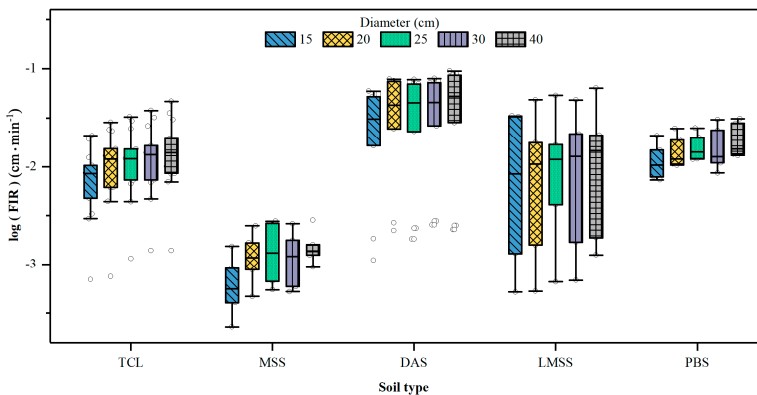

**Figure 5.** Distribution of the simulated log-transformed final infiltration rate (FIR) with different inner-ring diameters for five soil types.

The average FIT variation range of different soil types was relatively small, among which, FIT for MSS and DAS tended to increase slowly as the inner-ring diameter increased. FITs for TCL, LMSS, and PBS were higher in the infiltration process with an inner-ring diameter of 15 cm. Similar to Figure 5, LMSS had the longest box line and obvious polarization, indicating that LMSS contained various types of infiltration processes (Figure 6).

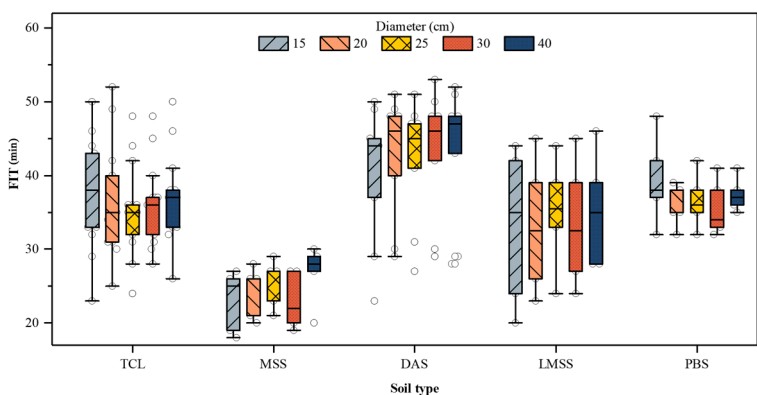

**Figure 6.** Distribution of the simulated final infiltration time (FIT) with different inner-ring diameters for five soil types.

The parameters and performance of the six infiltration models and the final infiltration parameters at each site show that the scale effect can be weakened by increasing the infiltrometer diameter. The infiltrometer with a 40 cm inner-ring diameter was less affected by the scale effect but not enough to completely overcome it. So far, there are few studies on the scale effect of double-ring infiltration in semi-arid steppes. A proper infiltrometer diameter not only satisfies the test accuracy but also reduces the test consumption. Studies showed that a small infiltrometer diameter leads to difficulty in soil

moisture transport (the infiltration ring with an inner-ring diameter of 15 cm in Figure 3). This was because when the infiltration ring penetrated into the soil, its ring was bound to cut the soil so that the soil squeezed to the inner and outer sides of the ring, resulting in an increase in the compactness of the soil within the infiltration ring, thus slowing down the infiltration rate [45]. As the infiltrometer diameter increased, the ratio of circumference to area decreased, and the influence of the ring on the soil also decreased. Therefore, appropriately increasing the infiltrometer diameter was more conducive to reducing the scale effect, but not the larger the better, since it would consume more water. It is not easy to get water in areas that are already short of water and have poor transportation. Carrying a large amount of test water and large test instruments is not convenient for field test transportation.

K-means clustering was adopted to regroup FIR and FIT, and the results are presented in Figure 7. Infiltration sites could be divided into three groups according to FIR and FIT. Sampling sites in group I were mainly distributed near the river. Their FIRs were slow and concentrated, with FIT generally within a short 30-minute period. Group III sites mainly included DAS, where FIRs were rapid. The FIR and FIT of the sites in Group II were between which in the Groups I and Group III, and more concentrated. It was found that the soil infiltration distribution in the study area was not consistent with the soil type distribution. Not all sites near the river with MSS and LMSS were distributed in group I, and all sites with DAS were in group III. The actual situation was that there were sampling sites close to the river where FIR was very slow. The sites away from the river, with low soil moisture content, had a quick FIR.

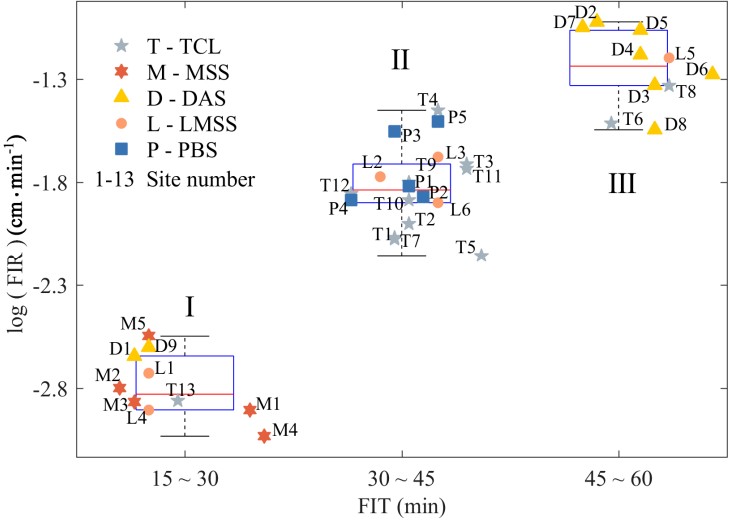

**Figure 7.** Regrouping of log-transformed final infiltration rate (FIR) and final infiltration time (FIT) at each infiltration site.

By comparing the FIR and FIT differences between the soil type map and the double-ring infiltration test results (Figure 8), we clearly found that there was basically no change in FIR and FIT in the eastern part of the study area where MSS and LMSS intersected. As the longitude moved to the west, the moisture decreased in time, and the FIR and FIT in the soil infiltration distribution map increased accordingly, where there was an extreme deviation from the predicted results based on the soil type map. FIR and FIT had a maximum reduction of 0.15 cm·min$^{-1}$ and 11 min. Compared with the predicted results from the soil type map, TCL and DAS similarly predicted an overestimation of 0.15–0.2 cm·min$^{-1}$ and 20 min in FIR and FIT near the river relative to the soil type map. DAS in the middle of the study area had a high similarity with the soil infiltration distribution map, while DAS in the northwest showed that the infiltration rate was slightly slower, but the FIT was basically the same. It can be seen that the soil infiltration distribution varied from the soil type distribution, and the soil type could not fully explain the infiltration process because the soil type was just one of the factors affecting the infiltration characteristics.

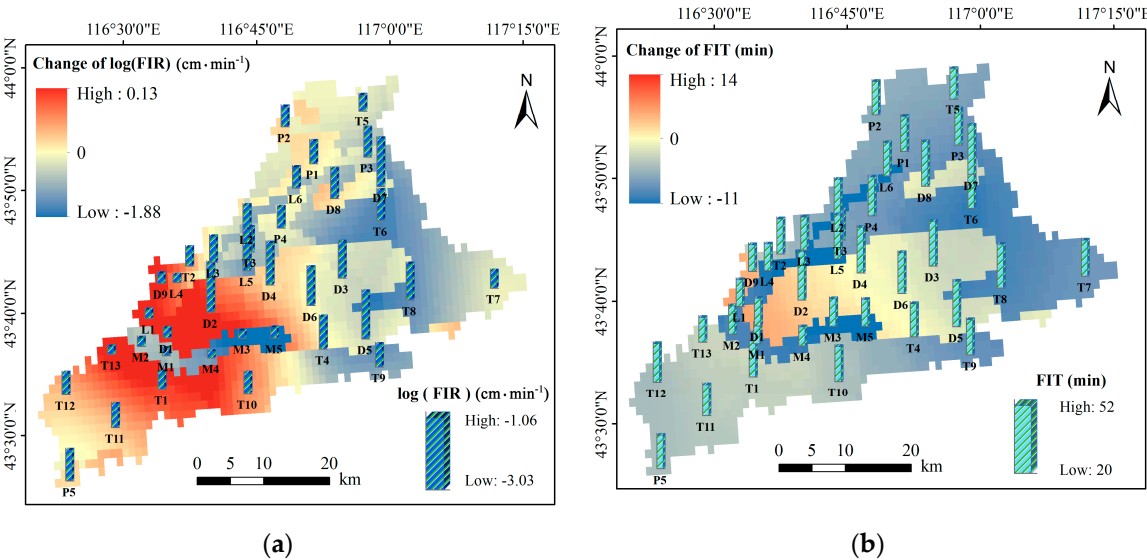

**Figure 8.** The change in the log-transformed final infiltration rate (FIR) (**a**) and final infiltration time (FIT) (**b**) between the soil type map and the experimental simulation map on the scale of 1 km × 1 km.

### 3.4. Soil Infiltration Map with Affecting Factors

PCA analysis was carried out on soil particle size, bulk density, organic matter content, initial moisture content, underground biomass, and water temperature at the 38 sampling sites. We obtained two principal components containing 77.22% of the original information. The study classified these influencing factors as soil physical properties and the external environmental components (Figure 9).

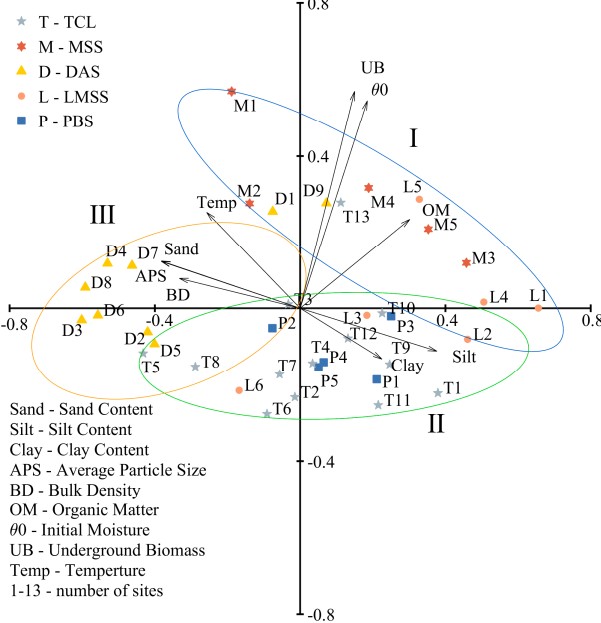

**Figure 9.** Principal component analysis (PCA) ordination with plotted soil and environmental variables.

PCA results showed that the three regions with K-means clustering classification could also be well-differentiated. The average particle size, which was located in the arrangement of group III in the left of the map, was larger than other soil properties. The average sand content in DAS was more than 94%, which led to a higher bulk density in the soil [46]. The test time for DAS was also basically at noon due to the location of the sampling site. A high water temperature can reduce the viscosity of water [47]. The combined effect of the two factors made the infiltration rate in this area relatively high.

Group I, with sites located near the river, was mainly distributed in the first and fourth quadrants of the permutation diagram. The average soil moisture content, which exceeded 26%, not only reduced the soil infiltration rate, but also promoted plant growth. As a result, the underground biomass and soil organic matter content were more than twice that of other arid regions. The root system and organic ingredients of the plants could make the soil particles bind more tightly and increase the soil pores. However, due to the lack of strontium, they accumulated, which reduced the movement of soil moisture to some extent. This was consistent with the views of Liu et al. [47] and Wang et al. [45].

The FIR using a double-ring infiltrometer was the FIR in unsaturated soil, which was typically smaller than a saturated soil FIR, which is consistent with the fact that the water passed through the belt of the pack [48]. The faster FIR showed that the soil water-holding capacity was poor. The stability of soil infiltration conditions and influencing factors were scale-dependent, that is, soil with different properties did not have the same infiltration characteristics on spatial and temporal scales. FIR could distinguish the infiltration characteristics on the spatial scale, while FIT measured the characteristics of this change well on the time scale. The decrease of infiltration rate with time was the macro expression of soil moisture gradually moving to an unsaturated and stable migration state. The faster the soil reaches this state, the more sufficient the conditions are for the soil to reach this state, that is, it has a higher water retention capacity [10]. This also has special circumstances, such as when the soil compaction is high, the soil wet front will move very slowly. In that case, the retention capacity is not good at all, the water does not infiltrate into the soil, and it will not be available for the plants, but will run off or become stuck on the surface and evaporate.

From the simulation parameters and performance changes of the infiltration models, it can be seen that the inner ring-diameter reached 40 cm and was not stable. In fact, the actual FIR was still larger than the $f_c$ simulated using Horton's model when the inner-ring diameter reached 40 cm. However, this does not mean that this study is of no practical value. On the one hand, we know that the inner-ring diameter of the double-ring infiltration test should be greater than 40 cm to minimize the scale effect in similar semi-arid grasslands. On the other hand, the research results show that, in the series of double-ring infiltration tests, the infiltrometers with an inner-ring diameter of 40 cm were subject to the minimum scale effect and the trend of this effect decreased with increasing inner-ring diameter. Therefore, the double-ring infiltration process with an inner-ring diameter of 40 cm and 0.33 outer-ring buffer index was still representative.

The whole study area had the feasibility of spatial interpolation, with few human interference factors and little difference in underlying surface conditions [49]. The infiltration area was divided into three parts (Figure 10): Area I: Valley wetland, close to the river, was affected by sufficient water and a large number of plants, with a slower infiltration rate and a better ecological environment. Area II: Pastoral grassland, where a large number of livestock were raised, and the soil infiltration rate was at a moderate level. Influential factors were complex, and the ecological environment had a certain degree of degradation. Area III: Desertification sandy land, with low vegetation coverage and serious ecological degradation, the infiltration rate was relatively fast under the influence of sandy soil, and the water-holding capacity of the soil was extremely poor. Through the interpolation expansion of the principal components in PCA, the arithmetic mean value of the first two principal components at each point was taken as the influential degree of soil properties and environmental factors on the soil infiltration process, and the depth of each region represented by the color was expressed. According to the quadrants of the load of each factor in PCA, the dominant factors influencing the infiltration process were represented. It could be seen that in the east area, especially area III and its surroundings, the soil texture was the dominant factor controlling the infiltration. The water-holding capacity of the soil in this region could be improved by adding organic carbon into the soil [10]. Area I and its surroundings in the west were more affected by external environmental factors. The reason for the good ecological environment and soil water-holding capacity of this region was that it had sufficient water, so ensuring the abundant water resources in this region was the key to maintaining the soil water-holding capacity. Factors affecting infiltration in area II were more complex since a large number

of animal husbandry and human activities had made the water-holding capacity tend to decline in the region. It is more important for soil restoration to formulate more reasonable animal husbandry policies and regulations as soon as possible to reduce the damage of grassland vegetation.

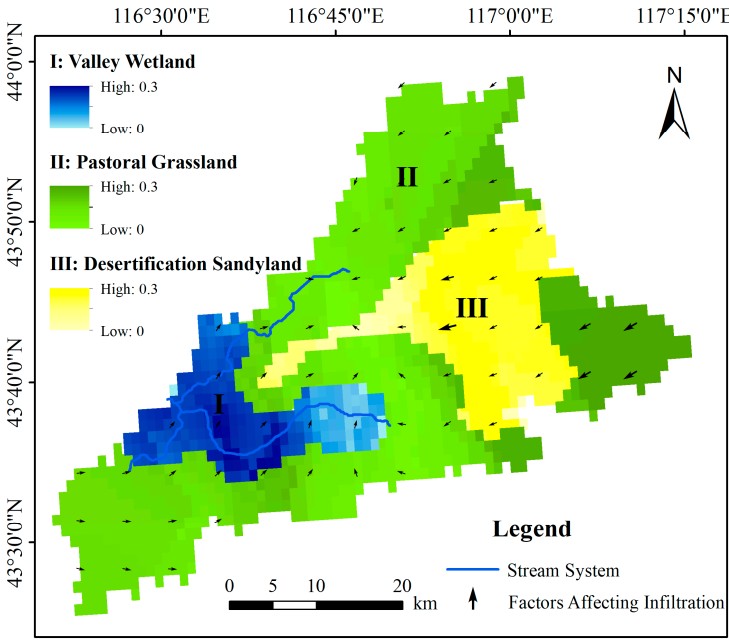

**Figure 10.** Map of the soil infiltration zones with the influence of soil and environmental factors, where the gradient color indicates the influence of soil properties and environmental factors on the infiltration process. The vector indicates the main influencing factors, where pointing to the first and third quadrants indicates that plants and moisture elements dominated, and pointing to the second and fourth quadrants indicates soil properties dominated.

## 4. Conclusions

In this study, 190 double-ring infiltration tests with different inner-ring diameters and the same outer-ring buffer index were carried out at 38 sites for five soil types in the Xilin river basin of a semi-arid steppe. The scale effect of the double-ring infiltration was compared using the parameters and model performance of six common infiltration models. The Horton model, which had the best simulation results and the 40 cm inner-ring diameter infiltration process with the smallest scale effect, was used to calculate FIR and FIT at each site to re-partition each test site and draw the soil infiltration map based on the characteristics of the infiltration process. The results showed that:

(1) Double-ring infiltration has a scale effect, which decreased with the increase of the inner-ring diameter. The infiltrometer with an inner-ring diameter of 40 cm could not completely overcome the scale effect.

(2) The model performance showed that the Kostiakov, Horton, Kostiakov-Lewis, and USDA-NRCS models were able to fit the infiltration process well in the semi-arid steppe.

(3) PCA analysis showed that the natural sandy meadow land in the study area was mainly affected by two factors: soil physical properties related to soil compactness and pore distribution, and external environmental components related to the kinetic energy potential of the infiltrating liquid.

(4) Rezoning based on infiltration characteristics could simplify the original soil type zoning and provide corresponding guiding suggestions for ecological restoration from the perspective of the soil.

To sum up, obtaining infiltration characteristics through a field infiltration test by drawing a zoning map is effective for gaining information to help with soil remediation. The method of selecting

the infiltrometer diameter and infiltration model could also be extended to the semi-arid steppe area of mid-latitude similar to the study area. However, further studies are needed to analyze the influencing factors, such as the soil compaction of some complicated soil and environmental infiltration processes, and to improve the use and improvement of soil infiltration zoning maps.

**Author Contributions:** M.L. developed the initial and final versions of this manuscript and analyzed the data. T.L., L.D., Y.L., and L.M. contributed their expertise and insights, overseeing all of the analysis and supporting the writing of the final manuscript. M.L., J.Z., Y.Z., and Z.C. performed the experiments.

**Funding:** This research was funded by the National Key R&D Program of China (No. 2018YFC0406400), the National Natural Science Foundation of China (Nos. 51620105003, 51139002, 51479086, and 51509131), the Inner Mongolia Natural Science Fund Key Project (No. 2018ZD05), the Ministry of Education Innovative Research Team (No. IRT_17R60), the Innovation Team in Priority Areas Accredited by the Ministry of Science and Technology (No. 2015RA4013), the Inner Mongolia Industrial Innovative Research Team (Grant 2012), the IMAU Innovative Research Team (Grant NDTD2010-6), and the program for Young Talents of Science and Technology in Universities of Inner Mongolia Autonomous Region (Grant NJYT-18-B11).

**Conflicts of Interest:** The authors declare no conflict of interest.

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
