# Peer review of "The Scale Effect of Double-Ring Infiltration and Soil Infiltration Zoning in a Semi-Arid Steppe"

_water, doi:10.3390/w11071457_

Round 1

Reviewer 1 Report

Please, refer to the attached file. Thanks.

Author Response

Dear reviewer,

Thank you very much for your consideration. We have revised our manuscript according to your comments and suggestions. We have point-by-point response to your comments in the upload Word file. Best wishes!

Kind regards,

Mingyang Li

Reviewer 2 Report

Dear Authors,

Congrtulation for the huge work done!!!

I made some suggestions in the text.

I am sure it will not take long to fix them.

Maybe you should consider mentioning in the title that the area was compacted and the steppe was a natural one but, I can live with the present title as well.

Best regards, Csabi

Author Response

(The authors gave the same response as above.)
